# Semi-CNN Architecture for Effective Spatio-Temporal Learning in Action Recognition

**Mei Chee Leong** [1] , **Dilip K. Prasad** [2,*] , **Yong Tsui Lee** [3] **and Feng Lin** [4]

[1] Institute for Media Innovation, Interdisciplinary Graduate School, Nanyang Technological University, Singapore 639798, Singapore; MLEONG006@e.ntu.edu.sg
[2] Department of Computer Science, UiT The Artic University of Norway, 9019 Tromsø, Norway
[3] School of Mechanical and Aerospace Engineering, Nanyang Technological University, Singapore 639798, Singapore; mytlee@ntu.edu.sg
[4] School of Computer Science and Engineering, Nanyang Technological University, Singapore 639798, Singapore; asflin@ntu.edu.sg
* Correspondence: dilip.prasad@uit.no

**Abstract:** This paper introduces a fusion convolutional architecture for efficient learning of spatio-temporal features in video action recognition. Unlike 2D convolutional neural networks (CNNs), 3D CNNs can be applied directly on consecutive frames to extract spatio-temporal features. The aim of this work is to fuse the convolution layers from 2D and 3D CNNs to allow temporal encoding with fewer parameters than 3D CNNs. We adopt transfer learning from pre-trained 2D CNNs for spatial extraction, followed by temporal encoding, before connecting to 3D convolution layers at the top of the architecture. We construct our fusion architecture, semi-CNN, based on three popular models: VGG-16, ResNets and DenseNets, and compare the performance with their corresponding 3D models. Our empirical results evaluated on the action recognition dataset UCF-101 demonstrate that our fusion of 1D, 2D and 3D convolutions outperforms its 3D model of the same depth, with fewer parameters and reduces overfitting. Our semi-CNN architecture achieved an average of 16–30% boost in the top-1 accuracy when evaluated on an input video of 16 frames.

**Keywords:** action recognition; spatio-temporal features; convolution network; transfer learning

---

## 1. Introduction

Action recognition via monocular video has valuable applications in surveillance, healthcare, sports science and entertainment. Deep learning methods such as convolutional neural network (CNN) [1] have demonstrated superior learning capabilities and potential in discovering underlying features when given a large number of training examples.

An action in video sequences can be characterized by its spatial and temporal features across consecutive frames. Spatial features provide contextual information and visual appearance of the content, while temporal features define the motion dynamics that happens in the range of the video frames. The task of action recognition is to effectively learn discriminative and robust spatio-temporal representations from video sequences for identifying different action classes. However, network performance often degrades when dealing with high variations of realistic and complex videos, due to major challenges such as occlusion, camera viewpoints, background clutter and variations in the subjects and motion involved.

Supervised learning with CNNs has been studied and exploited to perform action recognition, where representations in the spatial and temporal dimensions can be encoded in separate streams or simultaneously. Spatial features are extracted directly from RGB frames using 2D CNNs, while

temporal features are represented by pre-computed hand-crafted features such as optical flow or motion trajectory, or a stack of consecutive frames. Direct learning of spatio-temporal features from video frames can be implemented using 3D CNNs, which share a similar structure as 2D CNNs, but replace all the 2D convolution kernels with 3D ones.

This paper exploits the architecture of 2D and 3D CNNs and introduces an efficient fusion approach that combines the spatial layers in 2D CNN and spatio-temporal layers in 3D CNN. We utilize pre-trained models on ImageNet to initialize our 2D convolution layers and perform fine-tuning on the 1D and 3D convolution layers. Our empirical results demonstrate that segregation and fusion of convolution layers in the spatial and temporal spaces outperforms its 3D model of the same depth, when evaluated on the action recognition dataset UCF-101 [2].

## 2. Related Work

A two-stream architecture [3–5] consists of two separate 2D CNNs to train a classifier for each of the spatial and temporal features. The prediction scores from both streams will then be fused to form the final prediction. On the other hand, a 3D CNN [6–8] can directly learn spatio-temporal information when applied on consecutive frames, without explicitly computing the motion features. However, the number of trainable parameters increases substantially in deep models, leading to overfitting on smaller-scale datasets [9]. Different fusion methods have been investigated, such as fusing two-stream CNN and 3D CNN [10] or mixing 2D and 3D convolutions in an architecture [11], and splitting 3D convolution to 2D spatial and 1D temporal convolutions [12]. These methods have demonstrated improved performances over individual architectures that utilize full 2D or 3D convolutions.

**3D CNN.** In order to capture both spatial and temporal features across video frames, Ji et al. [6] proposed a 3D CNN model to perform 3D convolution on multiple frames and achieved better performance in action recognition as compared to 2D CNN. Jung et al. [13] improved the model by capturing multiple timescales at different layers of the convolutional network; while Tran et al. [7] experimented with different network architectures and kernel sizes for 3D convolutions. Hara et al. [8] built a 3D ResNets architecture, by exploiting the effectiveness of residual learning in deep 2D ResNets [14]. Varol et al. [15] trained 3D CNNs on both image frames and motion features using multiple spatio-temporal scales, to obtain combined results.

**Two-stream CNN.** Simonyan and Zisserman [3] designed a two-stream CNN architecture, which captures spatial and temporal features in separate CNNs before combining them to train a classifier. Gkioxari and Malik [5] trained separate CNNs for video frames and flow signals in region proposals for action prediction in individual frames. Instead of using the full image as input, their system search for region proposals and train the classifier for action prediction in individual frames. The identified action regions will be linked with consecutive frames to form connected motions. Similarly, Tu et al. [16] also employed a human-based region proposals to train a multi-stream CNN that consists of appearance, motion and region features as input stream. Zhang et al. [4] achieved real-time action recognition by learning motion vector from a pre-trained optical flow network. Diba et al. [17] encoded features from each CNN stream using temporal linear encoding and concatenate them to form a descriptor. Another work by Girdhar et al. [18] learned aggregation of features for both spatial and temporal streams, to form a new encoded representation in action classification.

**Spatio-temporal fusion.** Sun et al. [19] introduced the factorizing of 3D spatio-temporal convolution kernels to 2D and 1D kernels, where their architecture starts from 2D convolution layers and splits into two streams for spatial and temporal encoding. Feichtenhofer and Zisserman [10] introduced a fusion of two-stream CNN and 3D CNN, where fusion at different layers was investigated. Tran et al. [11] introduced two variants of spatio-temporal learning. The first one contains both 2D and 3D convolutions in a ResNet model, while the second variant decomposes a 3D convolution into a 2D spatial convolution and a 1D temporal convolution.

The variant of the architecture that is the most related to our proposed framework is the mixed convolutions network by Tran et al. [11]. The difference is that our architecture performs spatial

convolutions at the lower layers, followed by temporal convolutions and spatio-temporal convolutions at the higher layers. In addition, our model outperforms 3D ResNet by up to 18% in the top-1 accuracy for an input of 16 video clips when evaluated on the UCF-101 dataset, and it reduces overfitting issues as faced in 3D ResNet models [9].

## 3. Proposed Architecture

Our proposed framework started from adopting an existing CNN model as base model to initialize the network architecture. Pre-trained weights for 2D convolution layers were transferred to our semi-CNN to form the bottom layers of our architecture. Temporal convolution layers were then added to form the intermediate structure. Temporal encoding downsamples the depth features while preserving its spatial dimension. Lastly, spatio-temporal encoding layers were added on top before connecting to a fully-connected layer for classification prediction. Figure 1 illustrates the comparison of architecture designs for 2D CNN (Figure 1a), 3D CNN (Figure 1b) and our semi-CNN model (Figure 1c).

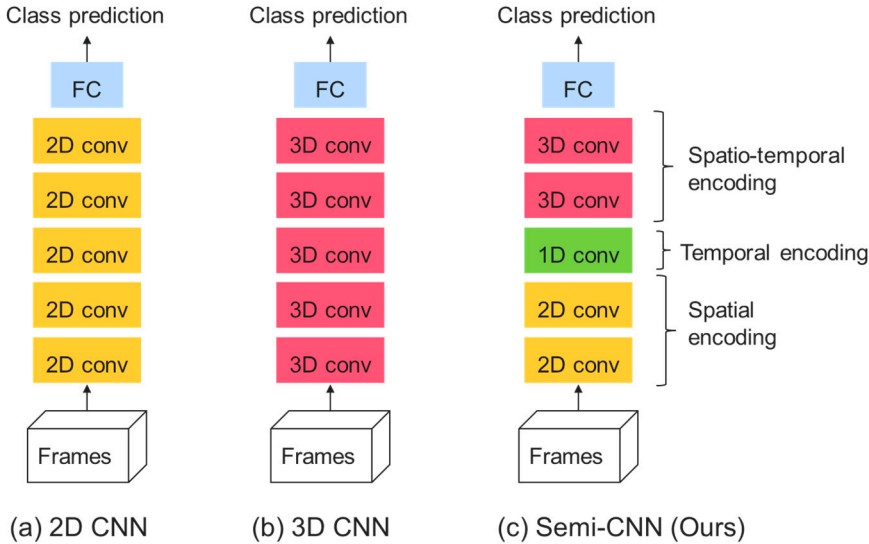

**Figure 1.** Architecture comparison for (**a**) 2D convolutional neural network (CNN), (**b**) 3D CNN and (**c**) our proposed semi-CNN.

### 3.1. Base Models

We constructed our architecture based on three popular convolutional networks: VGG-16 [20], ResNet [14] and DenseNet [21]. We attempted to preserve the network configurations and model's depth, while aggregating the layers to perform convolution in the spatial, temporal and spatio-temporal dimensions.

### 3.1.1. Very Deep Convolutional Networks (VGG)

The architecture of very deep CNN, named VGG, with a depth of 16–19 layers was introduced by Simonyan and Zisserman [20] for the task of large-scale image classification. The convolution filter size was fixed at $3 \times 3$ for all layers to allow deeper implementation while increasing non-linearity functions (by adding a rectification layer after each convolution) to learn complex representations. After the convolution layers, the output was max pooled before connecting to three fully-connected layers of 4096-D. This leads to a high number of learnable parameters, with 138 M for VGG-16 layers and 144 M for VGG-19 layers.

### 3.1.2. Residual Networks (ResNets)

When developing deeper networks by stacking more and more convolution layers, a critical problem arises during back-propagation where the gradients decrease and become too small or vanish while propagating to the lower layers. This prevents weights learning and updating in the layers, which prevents the network from converging. Hence, adding more layers without any remedy will result in stagnant accuracy or even poorer performance [14]. He et al. [14] proposed a residual learning framework, which incorporates a residual learning formulation in the stacked layers to allow better optimization. Residual learning is implemented using "shortcut connections" where the output of a previous layer can be mapped and added to the output of the connecting stacked layers. Another benefit of residual learning is that it does not incur additional complexity or parameters to the network, which makes the development of extremely deep network (up to 152 layers) possible, with complexity much lower than VGG nets.

### 3.1.3. Dense Convolutional Networks (DenseNets)

Another work proposed to build deeper architecture by connecting each convolution layer to all the subsequent layers, forming a densely connected network, DenseNet [21]. Feature maps from previous connected layers were re-used in subsequent layers, hence reducing the number of trainable parameters. Besides, the vanishing gradient problem is also addressed by the skip connections (similar to the shortcut connections in ResNets) between layers, which map the output of a previous layer to the next connecting layer, and allow the gradients from every layer to be accessed during back-propagation. DenseNet has a considerably lower number of parameters when compared with ResNet of the same depth, and yet retains a high training capacity when the model goes deeper.

### 3.2. Configuration Settings

Our semi-CNN performs spatial convolution first, followed by temporal convolution and finally spatio-temporal convolution. To retain the same network depth as its 2D convolution network, we reduced the number of layers in the spatial convolution blocks and added layers to the temporal and spatio-temporal blocks. When computing the spatial convolution, the output shape of the temporal depth was preserved, and similarly when computing the temporal convolution, the spatial dimension of the output was retained. During spatio-temporal convolution, features were convolved in all three dimensions. For semi-ResNets, residual learning was retained in all the convolutional blocks; while for semi-DenseNets, dense connections between layers in each block were preserved.

Our semi-CNN architecture took input with dimensions of $16 \times 224 \times 224$, where 16 denotes the number of consecutive frames, and $224 \times 224$ denotes the spatial dimension. The input would be passed through spatial convolution blocks and pooling layers to obtain spatial features with an output of size $16 \times 14 \times 14$. Then, the features were convolved with temporal blocks and downsampling to $8 \times 14 \times 14$. At the top layers of the architecture, we performed spatio-temporal convolution and pooling to reduce the feature size to $2 \times 3 \times 3$ or $2 \times 4 \times 4$, before connecting to a global pooling layer to obtain a 1-D representation. The representation would then be passed to fully-connected layers for classification prediction.

### 3.3. Comparison with 2D and 3D CNN

This section presents a comparison of the layers configuration in 2D, 3D and our semi-CNN, for models VGG, ResNet and DenseNet. The network configuration was simplified and presented in Table 1 for easy reference and comparison.

**Table 1.** Configuration comparison for 2D, 3D and semi-CNN architecture, using base model VGG, ResNet and DenseNet.

| Model | 2D-CNN | 3D-CNN | Semi-CNN |
|-------|--------|--------|----------|
| VGG-16 | 13 | 8 | (9, 1, 3) |
| ResNet-18 | BasicBlock [2, 2, 2, 2] | BasicBlock [2, 2, 2, 2] | BasicBlock ([2, 1], [1], [1, 1, 2]) |
| ResNet-34 | BasicBlock [3, 4, 6, 3] | BasicBlock [3, 4, 6, 3] | BasicBlock ([3, 2], [2], [3, 3, 3]) |
| ResNet-50 | Bottleneck [3, 4, 6, 3] | Bottleneck [3, 4, 6, 3] | Bottleneck ([3, 2], [2], [3, 3, 3]) |
| ResNet-101 | Bottleneck [3, 4, 23, 3] | Bottleneck [3, 4, 23, 3] | Bottleneck ([3, 2], [2], [12, 11, 3]) |
| ResNet-152 | Bottleneck [3, 8, 36, 3] | Bottleneck [3, 8, 36, 3] | Bottleneck ([3, 4], [4], [18, 18, 3]) |
| DenseNet-121 | [6, 12, 24, 16] | [6, 12, 24, 16] | ([6, 6], [6], [12, 12, 16]) |
| DenseNet-169 | [6, 12, 32, 32] | [6, 12, 32, 32] | ([6, 6], [6], [16, 16, 32]) |

The notation for the configuration setting for semi-CNN was set as ($[ss]$, $tt$, $[st]$), where $ss$ denotes the number of spatial layers with convolutional kernel ($1 \times x \times x$), $tt$ denotes the number of temporal encoding layers with kernel size ($x \times 1 \times 1$) and $st$ denotes the number of spatio-temporal layers with convolutional kernel size of ($x \times x \times x$). The value of convolutional kernel size, $x$, was set with reference to the configurations in the 2D base model. For instance, the spatial kernel size for semi-VGG-16 was ($1 \times 3 \times 3$), while for semi-ResNet-18, the kernel size for the first spatial convolutional block was ($1 \times 7 \times 7$). The notation [ . . . ] denotes a building block configuration that consists of a stack of convolutional layers. The value shown inside the bracket denotes the number of blocks constructed in the network. Both ResNet and DenseNet models were constructed with building blocks.

For VGG model, we compared with 2D VGG-16 [20], a 3D convolution network (C3D) [7] and our semi-VGG-16. C3D had 11 layers (eight spatial layers and three fully-connected layers), while VGG-16 had 16 layers (13 spatial layers and three fully connected layers) in their networks.

For ResNet, we compared with 2D ResNets [14], 3D ResNets [8] and our semi-ResNets. ResNet models were constructed with two types of residual blocks [14]—basic block and bottleneck block. Semi-ResNet was constructed with reference to the building blocks used in the 2D model.

Comparison of DenseNet is the 2D model [21], 3D model (derived from 2D network) and our semi-DenseNet model. Similarly, the network configuration is represented by a stack of building blocks that consists of densely connected layers. Dense connections between layers in each block were preserved in the semi-CNN architecture.

The configurations described in Table 1 were not optimized, but have shown reasonably good performance when validated in the experimental section. A detailed architecture comparison could be found in Table A1 in the Appendix A.

## 4. Experiments

### 4.1. Implementation Details

We evaluated our framework using the popular action recognition dataset, UCF-101. It contained a total of 13320 video clips, with 101 action classes, and was divided into three training–validation splits. In our experiments, we only utilized the split-1 training and validation sets for evaluation. During training, 16 consecutive frames were randomly sampled from each video. Input frames were re-scaled and randomly cropped at multiple scales, before resizing them to the size of $224 \times 224$. They were also randomly flipped horizontally to allow data augmentation for better training. The input was normalized using the mean and standard deviation of the ImageNet dataset. For validation, the input frames from each validation video were sampled at fixed locations. The frames were rescaled and center cropped, without horizontal flipping.

Our network was trained end-to-end using stochastic gradient descent with learning rate 0.1, momentum 0.9, dampening 0.9 and weight decay of 0.0001. The learning rate was reduced by a factor of 10 when the validation loss value did not decrease for 10 epochs. The network architecture and training was implemented in PyTorch with CUDA, utilizing two GPUs of GeForce RTX2080Ti. Due

to memory constraints, we implemented accumulated gradients during training to allow processing of large batch sizes, while retaining the size of the computation graph. We used a mini batch size of 8 (or 4 or 2 depending on the network complexity) and accumulated the gradients for 32 iterations (effective batch size is 256) before back-propagation. The architecture was trained for 50 epochs and we presented the validation results for comparison.

## 4.2. Trainable Parameters

This section presented the number of trainable parameters for each model VGG, ResNet and DenseNet, and compared with the architectures of full 2D, 3D and our semi-CNN model. As our model utilized transfer learning to initialize our network parameters, we also presented the number of pre-trained parameters for each network. The 2D convolution network took an input of size (48, 224, 224), and its input channel size was 48, while the input dimension for 3D and our semi-convolution was (3, 16, 224, 224), with channel size 3 for each frame's RGB.

Table 2 describes the number of training parameters for VGG, ResNet and DenseNet models. As can be seen from the table, 3D convolution networks had a much higher number of trainable parameters as compared to their corresponding 2D convolution networks. Our semi-CNN architecture had the lowest number of parameters for the VGG-16 model. As for ResNet and DenseNet models, our network had higher parameters than 2D convolution networks, but lower than 3D convolution networks. The trainable parameters were obtained from the lower layers of the pre-trained network for our spatial convolutions only.

**Table 2.** Number of training parameters in millions(M) for VGG, ResNet and DenseNet models.

| Model | 2D-CNN | 3D-CNN | Semi-CNN | |
|---|---|---|---|---|
| | Params | Params | Pre-Trained Params | Total Params |
| VGG-16 | 134.7 M | 179.1 M | 5.3 M | 82.2 M |
| ResNet-18 | 11.4 M | 33.3 M | 0.4 M | 31.7 M |
| ResNet-34 | 21.5 M | 63.6 M | 0.8 M | 60.5 M |
| ResNet-50 | 23.9 M | 46.4 M | 0.9 M | 45.8 M |
| ResNet-101 | 42.8 M | 85.5 M | 0.9 M | 84.8 M |
| ResNet-152 | 58.5 M | 117.6 M | 1.4 M | 115.6 M |
| DenseNet-121 | 7.2 M | 11.4 M | 0.8 M | 10.4 M |
| DenseNet-169 | 12.8 M | 18.8 M | 0.8 M | 17.9 M |

## 4.3. Evaluation on Validation Dataset

This section presented the evaluation results for our architecture and the comparison of its performance with the corresponding 3D-CNN model. Since a previous study [7] has demonstrated that 3D-CNN outperforms 2D-CNN in action recognition tasks, we did not explicitly train 2D-CNN models for our comparison here.

The validation results on the UCF-101 Split-1 dataset are presented in Table 3. These results were obtained after training for 50 epochs, which might not have fully converged, but were presented here for comparison and reference. These results show a significant improvement on the prediction accuracy for our semi-CNN when applied on all the different models—VGG, ResNet and DenseNet. For the VGG-16 model, our architecture outperforms C3D [7] by 18% in accuracy. For ResNet models, an average of 16% improvement was achieved, while for the Densenet-121 model, the boost was almost 30%. Deeper networks of ResNet and DenseNet (such as ResNet-101 and DenseNet-169) were not presented here as the validation results deteriorate for both 3D- and semi-CNN architectures, demonstrating that the UCF-101 dataset was too small to be trained for a deep 3D network, as stated in [9,22]. Figure 2 illustrates the comparison of the training performance for the models listed in Table 3.

**Table 3.** Comparison of validation results for models VGG, ResNet and DenseNet, with 3D-CNN and semi-CNN architectures.

| Model | 3D-CNN | | Semi-CNN | |
|---|---|---|---|---|
| | Top-1 acc (%) | Top-5 acc (%) | Top-1 acc (%) | Top-5 acc (%) |
| VGG-16 | 36.53 | 62.70 | 54.27 | 81.21 |
| ResNet-18 | 47.11 | 73.65 | 64.92 | 85.51 |
| ResNet-34 | 48.19 | 74.41 | 66.53 | 88.58 |
| ResNet-50 | 37.67 | 64.60 | 50.67 | 78.91 |
| DenseNet-121 | 12.64 | 37.88 | 42.48 | 76.53 |

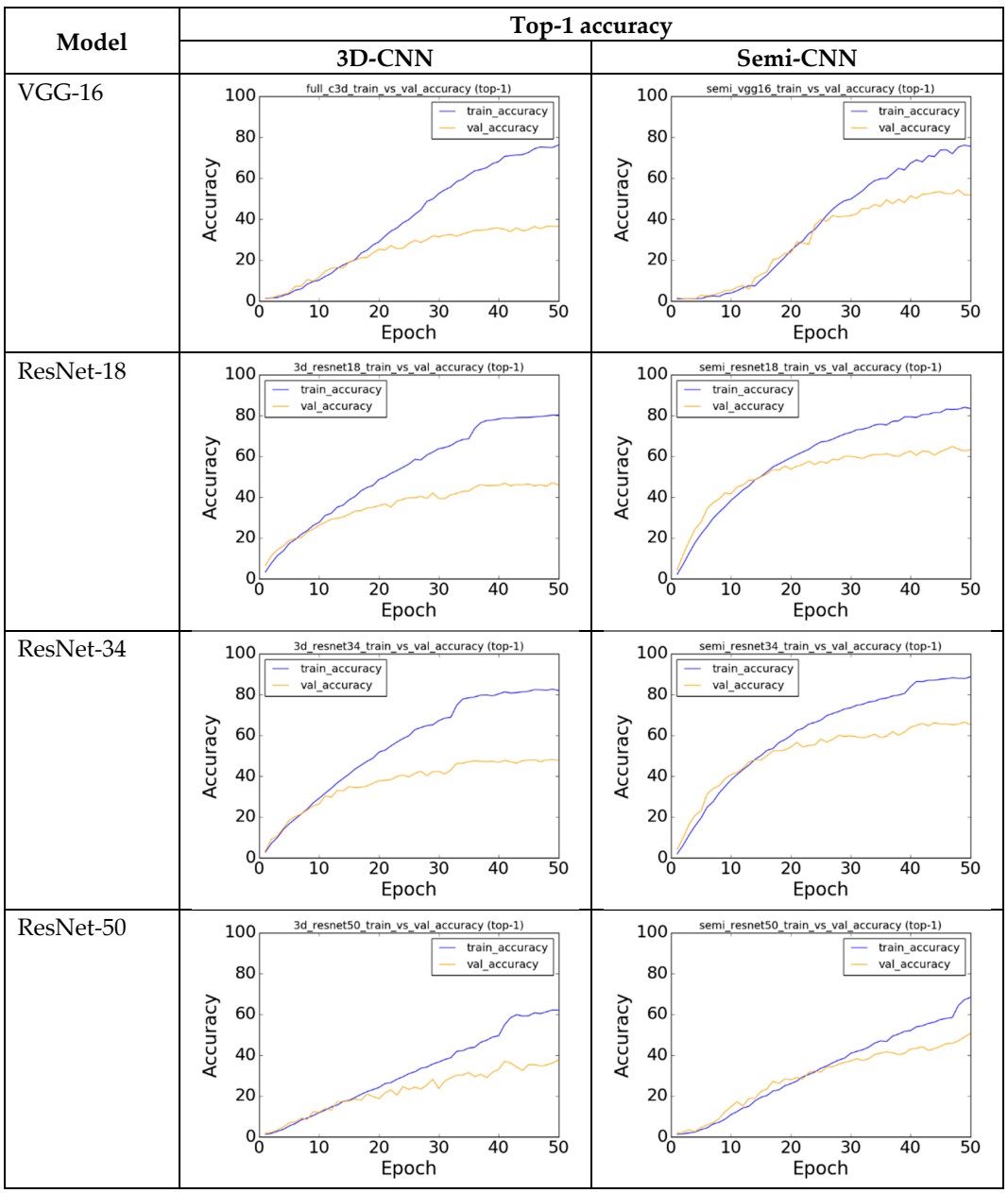

**Figure 2.** *Cont.*

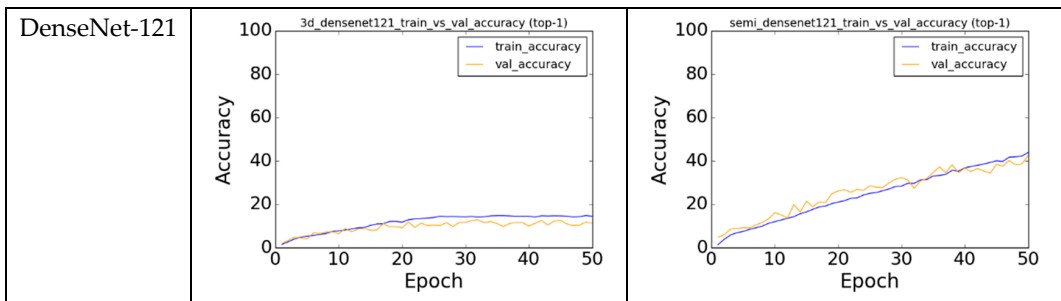

**Figure 2.** Performance plots for 3D-CNN and semi-CNN architectures applied on VGG, ResNet and DenseNet models.

In Figure 2, the gap between the training and the validation plots (blue and orange lines) for 3D-CNN models was much bigger as compared to the plots in semi-CNN. This indicates that overfitting occurs when we train a more complex model (or a deeper network) on a relatively small dataset. Our architecture learned faster as could be seen in the steeper slope, which is a result of transfer learning, which initializes the parameters in our spatial convolution layers. For deeper networks such as ResNet-50 and DenseNet-121, huge fluctuations could be seen in the validation loss of the 3D architecture, which could mean that the network is overfitting as it could not generalize its training parameters on the validation set. Although fluctuations also occur in our semi-CNN architecture, it is more stable and the validation loss is progressively reduced when trained for more epochs.

This empirical study provides strong support for our semi-CNN architecture in that a fusion of convolution layers outperforms a full 3D convolution network, with additional advantages on: (1) lower number of training parameters, (2) transfer learning from pre-trained models is feasible and (3) more stable training and faster convergence. Due to our limited computational resources, we only presented our validation results on shallower networks on UCF-101 dataset, and we expected that the performance could be generalized to deeper networks when applied on larger datasets.

### 4.4. Features Visualization and Qualitative Results Comparison

We reported the comparisons of qualitative results for 3D- and Semi-ResNet-18 models, as well as illustrations of features for Semi-ResNet-18 in Figure 3. The top two rows of Figure 3 show 16 consecutive frames sampled from the validation video as network input. Subsequent three rows in Figure 3 illustrate examples of features extracted in the spatial, temporal and spatio-temporal space in Semi-ResNet-18 (detailed architecture is presented in Table A1 in the Appendix A). Spatial features shown were obtained after the spatial convolution and pooling layers, with output dimension ($16 \times 14 \times 14$). After the temporal convolution block and max pooling layer, we obtained temporal features with dimension ($8 \times 14 \times 14$). The features were then processed by spatio-temporal blocks, where the output size was further reduced to ($2 \times 4 \times 4$). At the bottom row, we illustrated the top-5 prediction scores for both 3D- and Semi-ResNet-18.

Figure 3a–c show examples where our semi-CNN model outperformed 3D-ResNet, while Figure 3d shows example of 3D-ResNet with better performance. Figure 3e displays an example where both models make the correct prediction, while Figure 3f displays an example where both models fail. From the top-5 prediction scores in Figure 3, it is obvious that some of the predictions did not match with the action class. This could be due to: (1) the network having not fully converged and requiring more training epoch, and (2) insufficient training data to train the high number of network parameters.

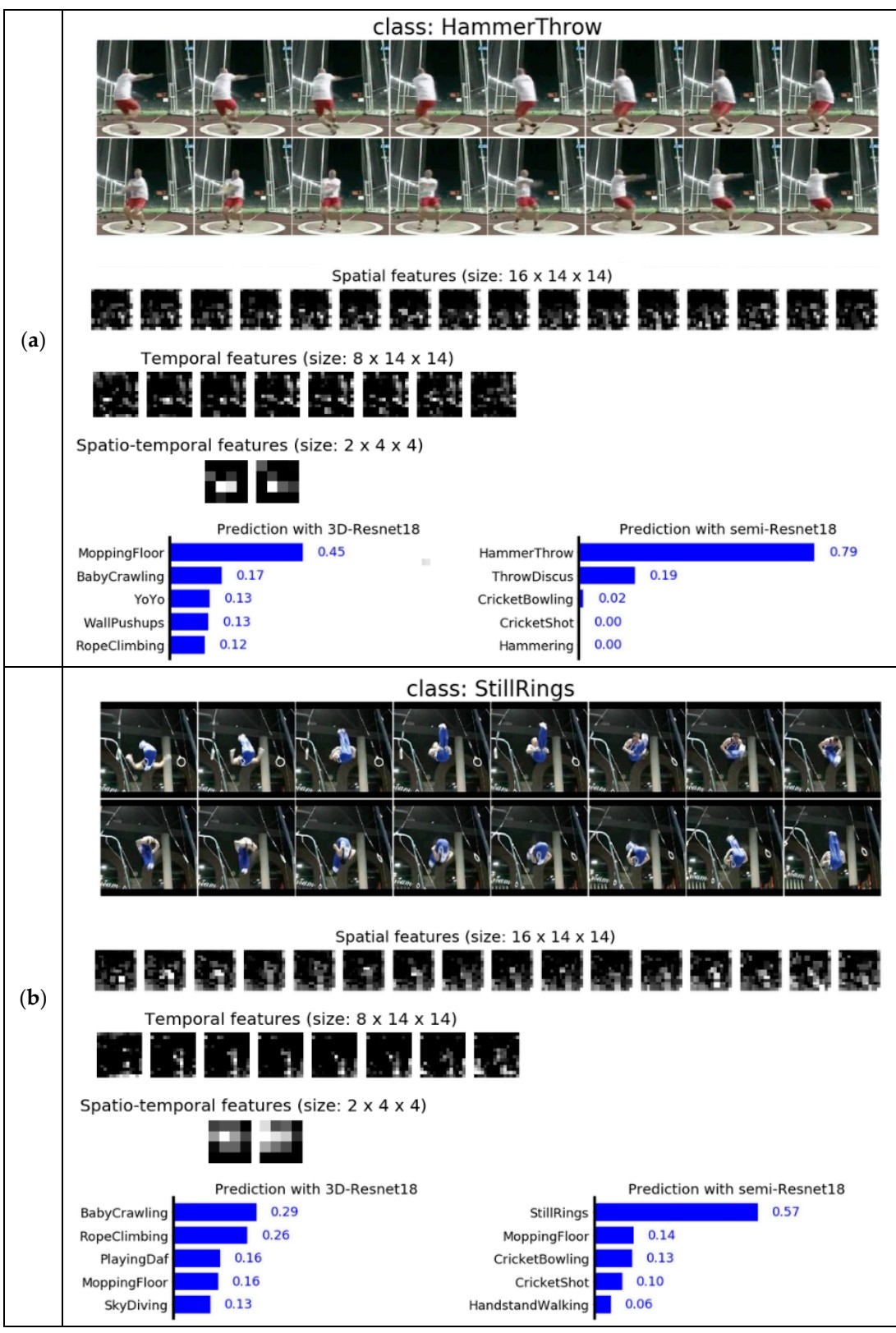

**Figure 3.** *Cont.*

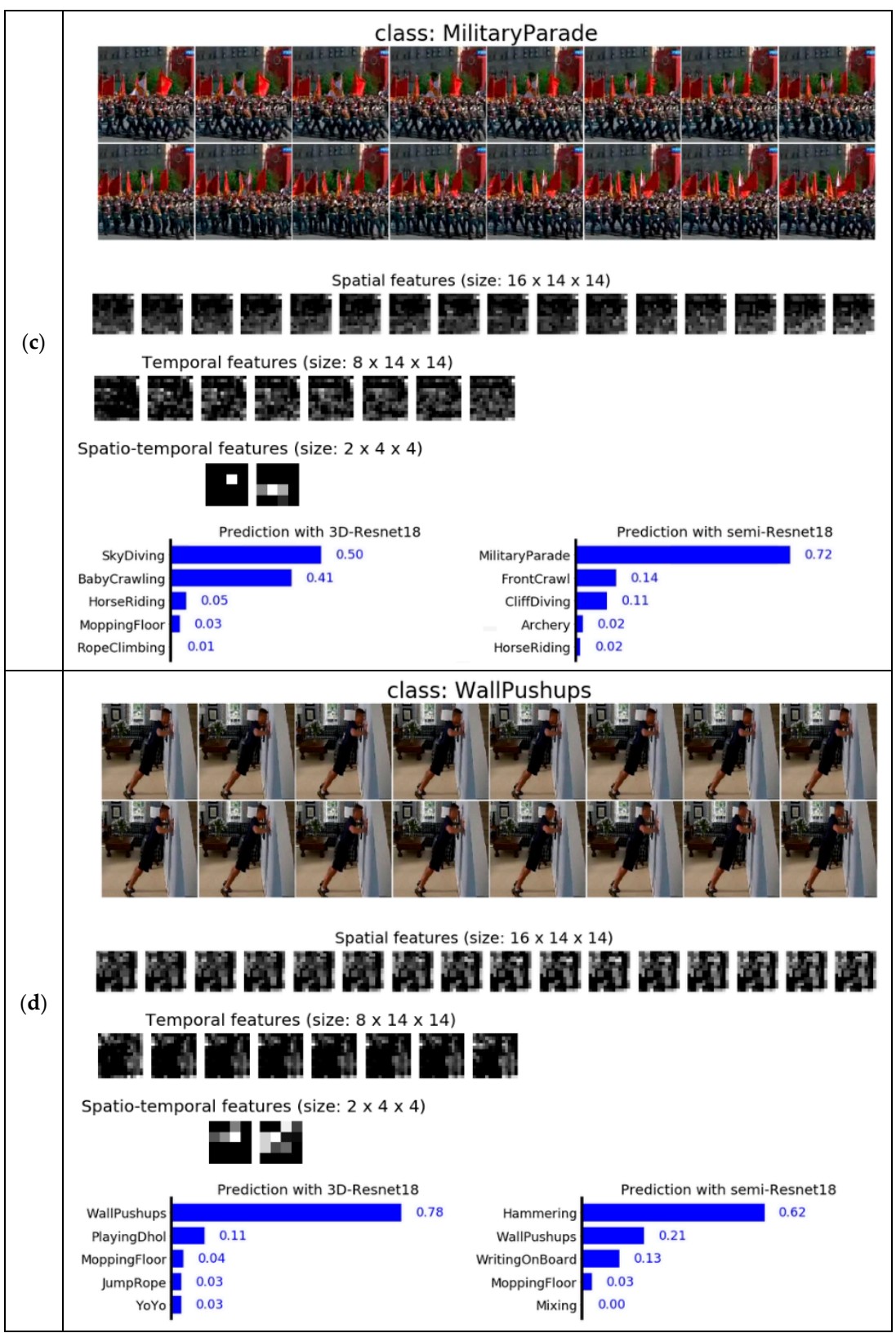

**Figure 3.** *Cont*.

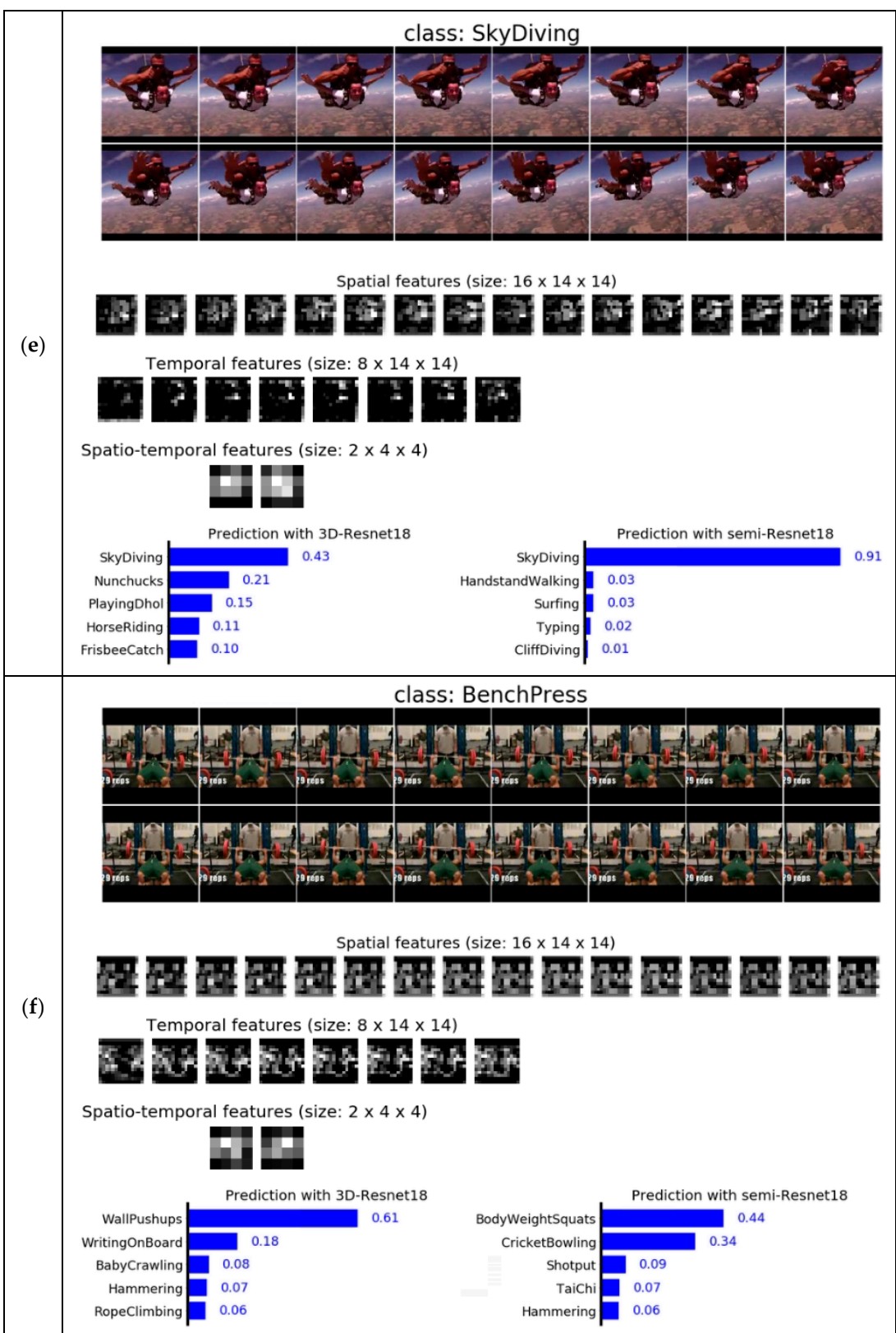

**Figure 3.** Examples of qualitative results for 3D- and Semi-ResNet-18 for different action classes: (**a**) hammer throw, (**b**) Still rings, (**c**) Military parade, (**d**) Wall pushups, (**e**) Sky diving, and (**f**) Bench press.

### 4.5. Experiment on Discontinued Motion

So far, the evaluation for semi-CNNs and 3D CNNs are based on continuous motion input with 16 consecutive frames. This section further investigated the network capability in learning spatio-temporal features when there exists motion discontinuity between frames. For experiment, we divided a full video into four segments and sample four consecutive frames from each segment to form the same input size of 16 frames. These $4 \times 4$ frames with motion discontinuity were then fed to CNN architectures for training. We utilized the same architecture and training parameters as described in Section 4.1 and reported the comparison results in Table 4.

**Table 4.** Comparison of validation results for discontinued motion input trained on 3D-CNN and semi-CNN architectures.

| Model | 3D-CNN | | Semi-CNN | |
|---|---|---|---|---|
| | Top-1 acc (%) | Top-5 acc (%) | Top-1 acc (%) | Top-5 acc (%) |
| VGG-16 | 42.61 | 70.18 | 57.94 | 82.58 |
| ResNet-18 | 47.87 | 75.23 | 68.65 | 88.79 |
| ResNet-34 | 47.45 | 73.83 | 70.50 | 89.00 |
| ResNet-50 | 34.05 | 63.20 | 53.77 | 80.62 |
| DenseNet-121 | 13.14 | 36.37 | 44.25 | 77.72 |

Semi-CNNs with frames sampled from different segments of a full video provides better action prediction than using a stack of consecutive frames. The average performance boost for semi-CNNs is 3.2% when compared to the results in Table 3. For 3D CNNs, the performance varies across models. VGG model shows an improvement of 6.08%, while the accuracy of ResNet-34 and ResNet-50 decreases. Do note that 3D-VGG-16 has more than double the number of training parameters as compared to Semi-VGG-16, but its accuracy is still much lower than our network's performance, with a difference of 15.3%. For ResNet models, Semi-ResNet consistently shows improvement of 3% even for deeper models, while the performances of 3D-ResNets deteriorate (from 0.8% to −3.6%) when the network goes deeper. For DenseNet-121, 3D-CNN does not show obvious improvement, with 0.5% increment, while semi-CNN shows an increment of 1.8% in accuracy.

This experiment has further validated the high learning capability of semi-CNN on spatio-temporal features, even when motion discontinuity occurs. This provides an advantage for video-level action recognition where motion changes in a full action can be captured for better discrimination.

## 5. Conclusions

The work in this paper demonstrated the effectiveness of our architecture as compared to existing fusion models and 3D convolution models. We evaluated our architecture on three popular models—VGG, ResNet and DenseNet. Our empirical results show significant improvements over the 3D convolution networks of the three models. In addition, our architecture shows faster convergence with transfer learning and had fewer training parameters, which reduced overfitting. The learning properties and the network depth for existing models were preserved. More experiments could be conducted to find the optimal configurations for the spatial, temporal and spatio-temporal convolution layers, as well as the segregation of the layers for each network. Deeper networks could be trained on larger datasets and fine-tuned on UCF-101 to further enhance the prediction accuracy.

**Author Contributions:** Conceptualization, methodology, validation, analysis and draft preparation, M.C.L.; software, resources, funding acquisition and draft review, D.K.P.; supervision, draft review and editing, Y.T.L.; draft review and editing, F.L. All authors have read and agreed to the published version of the manuscript.

**Funding:** The publication charges for this article have been funded by a grant from the publication fund of UiT The Arctic University of Norway.

**Conflicts of Interest:** The authors declare no conflict of interest.

# Appendix A

**Table A1.** Configuration comparison for 2D ResNet, 3D ResNet and our semi-ResNet, for an 18-layers network.

| Layer Name | 2D ResNet [14] | | 3D ResNet [8] | | Semi-ResNet (Ours) | |
|---|---|---|---|---|---|---|
| | 18-layer | Output Shape | 18-layer | Output Shape | 18-layer | Output Shape |
| Conv1 | $[7 \times 7, 64]$ stride 2 | (112, 112) | $[7 \times 7 \times 7, 64]$ stride (1, 2, 2) | (16, 112, 112) | $[1 \times 7 \times 7, 64]$ stride (1, 2, 2) | (16, 112, 112) |
| Max pool | $[3 \times 3]$ stride 2 | (56, 56) | $[3 \times 3 \times 3]$ stride 2 | (8, 56, 56) | $[1 \times 3 \times 3]$ stride (1, 2, 2) | (16, 56, 56) |
| Conv2_x | $\begin{bmatrix} 3 \times 3, 64 \\ 3 \times 3, 64 \end{bmatrix} \times 2$ | | $\begin{bmatrix} 3 \times 3 \times 3, 64 \\ 3 \times 3 \times 3, 64 \end{bmatrix} \times 2$ | | $\begin{bmatrix} 1 \times 3 \times 3, 64 \\ 1 \times 3 \times 3, 64 \end{bmatrix} \times 2$ | |
| Conv3_x | $\begin{bmatrix} 3 \times 3, 128 \\ 3 \times 3, 128 \end{bmatrix} \times 2$ | (28, 28) | $\begin{bmatrix} 3 \times 3 \times 3, 128 \\ 3 \times 3 \times 3, 128 \end{bmatrix} \times 2$ | (4, 28, 28) | $\begin{bmatrix} 3 \times 3 \times 3, 128 \\ 3 \times 3 \times 3, 128 \end{bmatrix}$ | (16, 28, 28) |
| Max pool | | - | | | $[1 \times 3 \times 3]$ stride (1, 2, 2) | (16, 14, 14) |
| Temporal conv block | | - | | | $\begin{bmatrix} 3 \times 1 \times 1, 128 \\ 3 \times 1 \times 1, 128 \end{bmatrix}$ | |
| Temporal max pool | | - | | | $[3 \times 1 \times 1]$ stride (2, 1, 1) | (8, 14, 14) |
| Spatio- temporal conv block | | - | | | $\begin{bmatrix} 3 \times 3 \times 3, 256 \\ 3 \times 3 \times 3, 256 \end{bmatrix}$ stride 1 | |
| Conv4_x | $\begin{bmatrix} 3 \times 3, 256 \\ 3 \times 3, 256 \end{bmatrix} \times 2$ | (14, 14) | $\begin{bmatrix} 3 \times 3 \times 3, 256 \\ 3 \times 3 \times 3, 256 \end{bmatrix} \times 2$ | (2, 14, 14) | $\begin{bmatrix} 3 \times 3 \times 3, 256 \\ 3 \times 3 \times 3, 256 \end{bmatrix}$ | (4, 7, 7) |
| Conv5_x | $\begin{bmatrix} 3 \times 3, 512 \\ 3 \times 3, 512 \end{bmatrix} \times 2$ | (7, 7) | $\begin{bmatrix} 3 \times 3 \times 3, 512 \\ 3 \times 3 \times 3, 512 \end{bmatrix} \times 2$ | (1, 7, 7) | $\begin{bmatrix} 3 \times 3 \times 3, 512 \\ 3 \times 3 \times 3, 512 \end{bmatrix} \times 2$ | (2, 4, 4) |
| Avg pool | $[7 \times 7]$ | (1, 1) | $[1 \times 7 \times 7]$ | (1, 1, 1) | $[2 \times 4 \times 4]$ | (1, 1, 1) |
| Fc | 101-d, softmax | | | | | |

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
