# Peer review of "Semi-CNN Architecture for Effective Spatio-Temporal Learning in Action Recognition"

_applsci, doi:10.3390/app10020557_

Round 1

Reviewer 1 Report

Action recognition has always been a topic of interest to the computer vision community, and increasing the accuracy of action classification has been considered a challenging task. This paper proposes an interesting 'semi-CNN' architecture that combines 1D, 2D, and 3D convolutions improving the action classification results compared to systems using pure 3D CNN-based architectures, reducing parameters to optimize, and lowering overfitting.

The results (both qualitative and quantitative) presented in the paper support the claims, and I believe this is an interesting architectural modification over 3D CNN-based action classifiers.

But, it would have been interesting to see:

The results of the proposed algorithm using other datasets More experiments for fine-tuning the network parameters, as mentioned in the conclusion

Author Response

Response to reviewer 1

Action recognition has always been a topic of interest to the computer vision community, and increasing the accuracy of action classification has been considered a challenging task. This paper proposes an interesting 'semi-CNN' architecture that combines 1D, 2D, and 3D convolutions improving the action classification results compared to systems using pure 3D CNN-based architectures, reducing parameters to optimize, and lowering overfitting.

The results (both qualitative and quantitative) presented in the paper support the claims, and I believe this is an interesting architectural modification over 3D CNN-based action classifiers.

Response: Thank you for appreciating the focus of our work and the results.

But, it would have been interesting to see:

The results of the proposed algorithm using other datasets More experiments for fine-tuning the network parameters, as mentioned in the conclusion.

Response: Thank you for useful suggestion to improve our work. To address more experiments concern we have added a new additional sub-section on 4.5. Experiment on Discontinued Motion. This section evaluates the same architectures with video input of discontinued motion changes, and presented the results in Table 4. Semi-CNN consistently achieved 3% improved performance across all the models, while some of the 3D-CNN performances deteriorate.

Reviewer 2 Report

Action recognition is an important topic and useful in many applications context.

The paper is well written and has a clear structure.

The important related work (3D CNN, two-streams CNNs, spatio-temporal fusion) is referenced in the paper.

The authors propose a efficient network architecture called Semi-CNN, that combines 2D CNN spatial layers and 3D CNN spatio-temporal layers, utilizing three different backbones (VGG, ResNet, DenseNet).

The experiments and evaluation on the validation dataset are well motivated and sound.
They show a clear advantage oft he Semi-CNN architecture with respect the the state of the art, as the number of model parameters could be reduced significantly.

Overall, this paper is a good contribution.

Author Response

Thank you reviewer for your time and effort in understanding our work and appreciating it.